# Effects of Remimazolam on Intracellular Calcium Dynamics in Myotubes Derived from Patients with Malignant Hyperthermia and Functional Analysis of Type 1 Ryanodine Receptor Gene Variants

**DOI:** 10.3390/genes14112009

**Published:** 2023-10-27

**Authors:** Hirotsugu Miyoshi, Sachiko Otsuki, Keiko Mukaida, Kenshiro Kido, Ayako Sumii, Tsuyoshi Ikeda, Yuko Noda, Toshimichi Yasuda, Soshi Narasaki, Takahiro Kato, Satoshi Kamiya, Yousuke T. Horikawa, Yasuo M. Tsutsumi

**Affiliations:** 1Department of Anesthesiology and Critical Care, Hiroshima University, Hiroshima 734-8551, Japan; sachi85@hiroshima-u.ac.jp (S.O.); mukaida.keiko1@gmail.com (K.M.); kidoken46@gmail.com (K.K.); batakobatake@gmail.com (A.S.); shiyotsudakei@gmail.com (T.I.); nodananoda0724@yahoo.co.jp (Y.N.); ijaran7@gmail.com (S.N.); takadai@hiroshima-u.ac.jp (T.K.); b054912@yahoo.co.jp (S.K.); yasuo223@hiroshima-u.ac.jp (Y.M.T.); 2Department of Anesthesiology, Hiroshima Prefectural Rehabilitation Center, Hiroshima 739-0036, Japan; toyasu@hiroshima-u.ac.jp; 3CHOC Health Alliance, Children’s Hospital Orange County, Orange, CA 92868, USA; yousuke.horikawa@gmail.com

**Keywords:** malignant hyperthermia, myotube, remimazolam, ryanodine receptor 1, intracellular calcium dynamics

## Abstract

Remimazolam is a novel general anesthetic and its safety in patients with malignant hyperthermia (MH) is unknown. We used myotubes derived from the skeletal muscle of patients with MH to examine the response to ryanodine receptor 1 (RYR1) agonist and remimazolam in MH-susceptible patients. Patients underwent muscle biopsy for the Ca^2+^-induced Ca^2+^ release (CICR) rate test, a diagnostic tool for MH in Japan. Ten patients had myotubes obtained from skeletal muscle cultures, and the genes associated with malignant hyperthermia in these patients were analyzed. The EC_50_ of caffeine, cresol, and remimazolam to induce intracellular calcium concentration change were compared between myotubes from CICR-negative genetic test patients and myotubes from other patients. Eight of the ten were CICR-positive, five of whom had RYR1 causative gene mutations or variants. Two patients had CICR-negative genetic tests, and as expected had the highest EC_50_ (the concentration of a drug that gives a half-maximal response) in response to caffeine, 4CmC and remimazolam. Three patients had a positive CICR but no known variants in RYR1 or CACNA1S (voltage-gated calcium channel subunit alpha1S). Myotubes in these patients had significantly lower EC50s for all agents than myotubes in CICR-negative patients. When myotubes from a patient who was CICR-negative and had no gene variant were used as a control, myotubes from CICR-positive patients were more hyper-responsive than controls to all stimulants used. The EC_50_ for remimazolam was lowest for myotubes from CICR-positive, RYR1-mutant patients, at 206 µM (corresponding to 123 µg/mL). The concentration was more than 80-times higher than the clinical concentration. *RYR1* gene variants in R4645Q and W5020G were shown to be causative gene mutations for MH. Intracellular calcium in myotubes from MH patients are elevated at high concentrations of remimazolam but not at clinically used concentrations of remimazolam. Remimazolam appears to be safe to use in patients with MH.

## 1. Introduction

Malignant hyperthermia (MH) is a hereditary muscle disorder induced via volatile inhalant anesthetics and depolarizing muscle relaxants used in general anesthesia [1]. The pathology of MH is abnormal hyperthermia and hypermetabolism caused by the dysregulation of intracellular calcium of type 1 ryanodine receptors (RYR1) in skeletal muscle [2]. The frequency of MH is about 1 per 100,000 patients with anesthesia. However, MH can still be a fatal complication as there is a possibility of death even when dantrolene is used at the time of onset [3,4]. Early administration of dantrolene is essential in the management of MH, and it is also important to avoid exposure in provoking anesthetics in patients predisposed to MH [5]. Thus, it is important to detect carriers with a genetic predisposition to MH prior to anesthesia, and to use anesthetics that do not induce MH in this population [6]. Dihydropyridine receptor (DHPR) is a receptor present in the cell membrane of skeletal muscle and functions as a voltage-gated calcium channel. Intracellular calcium is regulated by opening and closing of DHPR. This receptor is known as an MH-related receptor other than RYR1 [2].

In addition to the muscle contracture test, which is the gold standard, genetic testing is also widely used to detect predisposition to MH [7]. However, MH is not a disorder caused by a single *RYR1* gene mutation, but various *RYR1* gene mutations. Furthermore, *RYR1* genetic mutations may also be involved in other muscle diseases. Therefore, if a variant is found in RYR1 via genetic testing, we need to clarify whether the variant is pathogenic for MH through functional analysis. Myotube analysis is one of the methods to determine the predisposition to MH [8].

Intravenous anesthetics do not appear to induce MH because most MH results from exposure of predisposed individuals to volatile inhaled anesthetics [9]. Furthermore, the British Journal of Anaesthesia Education and the Malignant Hyperthermia Association of the United States (MHAUS) recognize that intravenous anesthetic does not induce MH [10,11]. The following are listed as intravenous anesthetics that can be safely used in MH patients: diazepam, etomidate, hexobarbital, ketamine, methohexital, midazolam, pentobarbital, propofol, and thiopental. Remimazolam is a benzodiazepine intravenous anesthetic. As the benzodiazepine intravenous anesthetic midazolam is safe for use in patients with MH, remimazolam may be similarly safe for use in patients with MH. On the other hand, as there are limited reports on remimazolam, a new general anesthetic, it is unclear whether remimazolam induces MH. A case report describes that a patient who was clinically diagnosed with MH during a previous surgery did not develop MH after receiving general anesthesia with remimazolam [12]. However, since the patient’s genetic information was not clear in this case report, it is unclear whether remimazolam is safe for patients with MH-related gene mutations. We have previously reported that exposure to remimazolam increased intracellular calcium in cultured cells. We have shown that the remimazolam-induced increase in intracellular calcium involves the release of calcium from the sarcoplasmic reticulum. That remimazolam-induced increase in intracellular calcium is not inhibited by the RYR1 antagonist dantrolene, intracellular calcium elevation by remimazolam did not change with or without RYR1 [13,14]. However, to further understand the effect of remimazolam on intracellular calcium dynamics, using cells that are closer to the cellular environment of patients with MH is essential. In particular, it is important to conduct research using samples for which the presence or absence of RYR1 and other genetic mutations related to malignant hyperthermia has been determined through genetic testing.

In the present study, we used myotubes derived from the skeletal muscle of patients with MH, which are closer to the human cellular environment than general cell cultures, to examine the response to RYR1 agonist and remimazolam in MH-susceptible patients.

## 2. Materials and Methods

### 2.1. Ethical Considerations

The study was performed in accordance with the Declaration of Helsinki and approved by the ethics committee of Hiroshima University (Epidemiology-68-6, HI-151-11) and written informed consent was obtained from patients. This study included minors amongst the subjects, and informed consent was obtained from all subjects and their legal guardians.

### 2.2. Patient Selection

Ten patients underwent muscle biopsy for the Ca^2+^-induced Ca^2+^ release (CICR) rate test, a diagnostic tool for MH [15]. Myotubes were obtained from skeletal muscle. Testing of the *RYR1* gene, the voltage-gated calcium channel subunit alpha1S (*CACNA1S*) gene, the gene for DHPR, and the SH3 and cysteine-rich domain 3 (*STAC3*) gene, was performed. Myotubes from one patient with a negative CICR test and no genetic variant was used as a control and compared with myotubes from the other nine patients.

### 2.3. CICR Rate Test

We measured the CICR rate using the method of Endo et al., which was previously reported to be an effective diagnostic tool for MH [15]. In brief, a muscle biopsy was obtained and dissected into a small bundle. The bundle was then prepared into skinned fibers. Fibers were then cut to 2–3 mm in length under a microscope and treated with various concentrations of Ca^2+^ (0, 0.3, 1.0, 3.0, and 10.0 mM). The isometric tension was measured using a force transducer to determine the CICR rate. MH predisposition was achieved if CICR was increased and defined as a CICR rate > two standard deviations (SD) above the mean of the control (IVCT and CHCT negative) [16] CICR values [17,18].

### 2.4. Gene Testing

DNA was extracted from specimens collected from patients, and a genetic analysis of RYR1, CACNA1S, and Stac3 was performed via gene panel sequencing. We extracted genomic DNA from patient blood or muscle fragments using the Dneasy Blood & Tissue Kit (QIAGEN, Hilden, Germany), according to the manufacturer’s instructions. Probes were designed using the xGen hybridization capture (Integrated DNA Technologies: IDT) of DNA libraries protocol. Libraries were prepared using the Lotus DNA Library Prep Kit (IDT) and xGen™ DNA EZ Library Prep Kit (IDT), and the xGen hybridization capture (IDT) of DNA libraries protocol. Libraries were prepared by the capture of DNA libraries using the xGen™ hybridization capture (IDT) of DNA libraries protocol. Sequencing was carried out with an Illumina NovaSeq 6000 platform using 150bp pair-end configuration. Germline variants were called with the Genome Analysis Toolkit Haplotypecaller, and Annovar v2019, 24 October, was used for variant annotation. Variant filtration followed GATK best practices.

The gene variants were included in the analysis using two or more of the in silico pathogenicity prediction programs, MutationTaster, CADD (combined annotation dependent depletion, score > 20), REVEL (rare exome variant ensemble learner, score > 0.5), SIFT (sorting intolerant from tolerant), PolyPhen-2, and less than 0.1% of minor allele frequency (gnomAD or ExAC). In addition, we examined the pathogenicity determinations at VCEP (ClinGen Variant Curation Expert Panel) guidelines and the European Malignant Hyperthermia Group (EMHG) scoring matrix for the variants.

### 2.5. Culture of Myotubes

Previously, we isolated myoblasts as described in [18]. In brief, myoblasts were obtained from skeletal muscle for CICR examination and cultured. Myoblasts were plated on 35-mm glass-bottom culture dishes with cell media supplemented with 10% FBS at 37 °C under 5% CO_2_. At 80% confluence, the medium was changed to cell media supplemented with 2% FBS to induce differentiation into myotubes. Seven to fourteen days after the medium change, measurements were performed on cells that had morphologically changed into multinucleated and spindle-shaped cells.

### 2.6. Ca^2+^ Fluorescence Measurements

Ca^2+^ imaging from a single myotube was performed using the Ca^2+^ fluorescence indicator, Fura-2/AM (Dojindo Molecular Technologies, Tokyo, Japan), as previously described [19]. In brief, cells were loaded and washed in Hank’s balanced salt solution (HBSS). Myotubes were perfused with HBSS for 30 min at a rate of 1.2 mL/min at 37 °C before the experiments. The myotubes were excited at 340 and 380 nm at 37 °C, and the measurements to evaluate intracellular Ca^2+^ changes were obtained at 510 nm using a fluorescence microscope (Nikon, Tokyo, Japan). Perfusion with HBSS was performed, and a Ca^2+^ image analyzer (Aquacosmos 2.5; Hamamatsu Photonics, Japan) was used to calculate the 340/380-nm wavelength ratio. Myotubes responding to 10 mM caffeine were included in the measurement [20].

### 2.7. Stimulant Loading Protocol

As we have previously reported, we used known stimulants of RYR1, caffeine, and 4-chloro-m-cresol (4CmC) [19,20]. Additionally, in this study we used remimazolam as a stimulant. Briefly, myotube viability was confirmed by testing the cell with 10 mM caffeine before measurements were obtained. The 340/380-nm wavelength ratio was measured when HBSS contained incremental levels of 4CmC. Each solution was washed out for 2 min before adding the next incremental level of 4CmC. 4CmC was tested at 3, 10, 30, 100, 300, 500, and 1000 µM. The sensitivity of caffeine was measured in the same way. Various concentrations of caffeine (0.25, 0.5, 1.0, 2.5, 5.0, 10.0, and 20.0 mM) were added to HBSS, and caffeine-induced changes in the 340/380-nm wavelength ratio were measured. Finally, remimazolam was used at multiple concentrations (10, 50, 100, 250, 500, 1000, and 1500 mM).

Measurements of intracellular calcium concentration changes in response to caffeine, 4CmC, and remimazolam were performed using different myotubes. Additionally, for each drug, one myotube was used per measurement.

### 2.8. Statistical Analyses

To trace concentration–response curves for caffeine, 4CmC, and remimazolam, the data were normalized to the maximal response of each myotube. Then, the half-maximal effective concentration (EC_50_), i.e., the concentration of stimulant required to reach half-maximal activation, was calculated from the acquired concentration–response curves. All data were analyzed using PRISM 9.2.0 software (GraphPad Software, San Diego, CA, USA). We compared the calculated EC_50_ between myotubes from control patients and myotubes from other patients in each individual patient using a one-way ANOVA with a Dunnett’s multiple comparisons test and; *p* < 0.01 was considered significant.

## 3. Results

### 3.1. Patient Background

There were 10 patients in total. Four patients had been diagnosed with MH, five patients had family members who had developed MH, and one patient was suspected of having MH after surgery. Six patients were male and four female. The youngest patient was 14 years old and the oldest patient was 80. The median age was 38 years old (Table 1).

### 3.2. Details of Variants Found by Genetic Testing

Details of in silico pathogenicity prediction programs for several variants of RYR1 and CACNA1S uncovered by genetic analysis of patients are shown in Table 2. We found two RYR1 causative gene mutations, three RYR1 variants, and one CACNA1S variant, as determined by the EMHG scoring matrix (Table 2).

All five causative gene mutations or variants found in RYR1 were included in the analysis, and the variant found in CACNA1S was not included in the analysis.

SNP: single nucleotide polymorphism; CADD: combined annotation-dependent depletion; REVEL: rare exome variant ensemble learner; SIFT: sorting intolerant from tolerant; VCEP: ClinGen Variant Curation Expert Panel; VUS: variant of unknown significance.

### 3.3. Morphological Change of Myotube

Myotubes were successfully isolated from all patients’ skeletal muscle (Figure 1). We did not see any significant morphological differences between the cultures. Cultured cells were tested between 7 and 14 days when myotube formation was identified. Cell viability was confirmed with its response to caffeine prior to any further testing of the cell.

### 3.4. Typical Intracellular Calcium Concentration Change

Figure 2A shows a typical increase in intracellular calcium concentration change upon exposure of the myotube to remimazolam according to the dosing protocol. Figure 2B shows concentration–response curves for the remimazolam and intracellular calcium concentration change (Figure 2).

### 3.5. Remimazolam Has Similar Effects on EC_50_ to Caffeine and 4CmC

Two patients had negative CICR and as expected had the highest EC_50_ in response to caffeine, 4CmC, and remimazolam. One of these patients had a CACNA1S variant (I639V) and was diagnosed as a variant of unknown significance (VUS) using VCEP guidelines and the EMHG scoring matrix. Three patients had a positive CICR but no known variants in RYR1 or CACNA1S. These patients had significantly lower EC_50_s in all three groups. Finally, five patients had known causative gene mutations or variants in RYR1 and the lowest EC_50_s amongst all the patients tested. *p* values were calculated using Patient 1, who was CICR-negative and had no gene variants in RYR1 and CACNA1S, as a control (Table 3).

## 4. Discussion

Remimazolam is a benzodiazepine anesthetic that is widely used worldwide as a sedative or general anesthetic. Remimazolam is a short-acting sedative with excellent concentration controllability, hemodynamic stability, and antagonistic properties [21]. On the other hand, it is speculated that remimazolam can be used safely in persons predisposed to MH, but the safety of remimazolam in MH patients has not been well studied [14]. Since myotubes can reproduce the cellular environment that reflects the genetic characteristics of patients with MH, it is possible to investigate responsiveness to drugs under conditions similar to the cellular environment of patients with MH. We investigated changes in intracellular calcium concentration in myotube cells from MH patients exposed to remimazolam [22]. Our results show that intracellular calcium did not increase with low concentrations of remimazolam, but intracellular calcium increased with high concentrations of remimazolam in any myotube. Additionally, the EC_50_ of remimazolam in myotubes ranged from 626.9 to 865.1 µM in those without a predisposition to MH (CICR test negative) and from 206 to 546.8 µM in those with a predisposition to MH (CICR test positive). Our study protocol exposed remimazolam at concentrations ranging from a minimum of 10 µM (corresponding to 6.0 µg/mL) to a maximum of 1500 µM (corresponding to 896 µg/mL). No increase in intracellular calcium concentration change was observed with any myotube, even at the lowest exposure. The EC_50_ of remimazolam blood-concentration required for general anesthesia in healthy adults is about 1.5 µg/mL [23]. Based on this, we considered that remimazolam at the clinically used concentration does not cause an increase in intracellular calcium. Furthermore, the EC_50_ for remimazolam was lowest for myotubes from CICR-positive RYR1-mutant patients, at 206 µM (corresponding to 123 µg/mL). Even in this case, the concentration was more than 80-times higher than the clinical concentration. Based on these findings, we thought that remimazolam at the clinically used concentration would not cause an increase in intracellular calcium concentration, even in patients with MH. Regarding the EC_50_ of remimazolam in myotubes, although there was a statistically significant difference between the EC_50_ of Patients 1 and 8, their dose–response curves were similar. On the other hand, Patient 8 clinically developed MH and also had a genetic mutation in RYR1 that causes MH. This suggests the possibility that the onset of MH cannot be predicted from the remimazolam reactivity of myotubes.

Our results show that remimazolam increased myotube intracellular calcium at lower levels in cells from patients diagnosed with MH predisposition using the CICR test than in cells from patients without MH predisposition. This indicates that myotubes from MH carriers increase intracellular calcium, even at low concentrations of the stimulant. This is consistent with previous studies using myotubes from MH-predisposed individuals. Similar results have been reported for the RYR1 stimulants, caffeine, and 4CmC, as well as the anesthetic, propofol [24,25,26]. Regarding the regulation of intracellular calcium, Migita et al., reported that high concentrations of propofol inhibit the uptake of intracellular calcium into the SR [24]. Furthermore, Fruen et al. reported that sarcoplasmic reticulum (SR) Ca^2+^-adenosine triphosphatase (Ca^2+^-ATPase) was inhibited by high concentrations of propofol [27]. In our study, a similar trend was observed for remimazolam in addition to caffeine and 4CmC. The increase in intracellular calcium in myotubes is considered to be also likely to increase in MH-predisposed subjects. Although the mechanism by which remimazolam regulates intracellular calcium is not clear, it is possible that remimazolam also inhibits the uptake of intracellular calcium into the SR by Ca^2+^-ATPase produced by high concentrations of propofol. On the other hand, it has been reported that the increase in intracellular calcium by remimazolam is not mediated by RYR1 [13,14]. In our study, intracellular calcium was also elevated in myotubes derived from patients without the *RYR1* gene variant. We believe that the responsive increase in intracellular calcium in myotubes derived from remimazolam-induced MH-predisposed patients indicates that these cells tend to increase intracellular calcium in general rather than the RYR1-mediated response.

Due to the spread of genetic testing, many gene mutations have been found in patients with MH [28]. Not all gene mutations found in MH patients are causative of MH; functional analysis of the discovered variants is also important [29]. RYR1, CACNA1S, and Stac3 are known to be causative genes for MH, and mutations in these genes are identified in 30–70% of patients with MH by genetic testing [30,31]. In other words, approximately 50% of the patients may have MH from an unknown mechanism. We performed functional analysis using myotubes in 10 CICR patients in our study. Eight of the ten patients were diagnosed with a predisposition to MH because they were positive for CICR. Three of the eight had no gene variants in any of the causative genes of MH, RYR1, CACNA1S, and Stac3. Among the 5 RYR1 causative gene mutations or variants found in our study, R163C and R2508H are gene mutations already listed as MH causative genes in EMHG. Nakano et al., analyzed the function of Q155K using rabbit RYR1 (Q156 K) and reported that it is hyperactive in MH [32]. Based on our report, we believe that Q155K satisfies the criteria for the causative gene of EMHG MH. On the other hand, R4645Q and W5020G are *RYR1* gene-variants whose functional analysis have not been reported in the past. We performed a functional analysis using caffeine and 4CmC as agonists and found that both R4645Q and W5020G have hyperactivity in MH. According to the EMHG criteria, ex vivo functional tests must be performed in two independent families. So, it is not possible to conclude that these genes are the causative genes of MH based on this report alone [10]. In order to conclude that these gene variants are the causative genes of MH, future reports of hyperfunction from other families are necessary. Interestingly, none of the three individuals (Patients 3–5) diagnosed with a predisposition to MH by CICR positivity had any genetic variants in RYR1, CACNA1S, or Stac3. These patients probably have MH caused by some unknown abnormalities in the regulation of intracellular calcium, even without gene variants in RYR1, CACNA1S, or Stac3. This indicates that MH predisposition diagnosis by CICR can determine the predisposition to MH even when there are no gene variants in RYR1, CACNA1S, or Stac3. Alternatively, this may be related to the specificity of MH predisposition-diagnosis in CICR.

Our study utilized patient-derived myotube cultures. However, these types of cell cultures have limitations in that they are not widely available and are unique to an individual patient and may affect our overall analysis. We were using samples from a patient with a negative CICR test and no genetic variants in RYR1 or CACNA1S detected by genetic testing as controls for comparison. Therefore, guarantees regarding the influence of unknown genetic variants and universality as a sample control may not be sufficient. This point is a limitation of our study. Next, we investigated remimazolam responsiveness to myotubes with several RYR1 variants. However, since MH occurs in association with multiple genetic mutations in RYR1 or CACNA1S, this result may not apply to all MH-associated genetic mutations.

## 5. Conclusions

We investigated the intracellular calcium responsiveness of myotubes to caffeine, 4CmC, and remimazolam. Myotubes derived from patients with MH increased intracellular calcium levels at high concentrations of remimazolam, but clinically used concentrations of remimazolam did not increase intracellular calcium.

## Figures and Tables

**Figure 1 genes-14-02009-f001:**
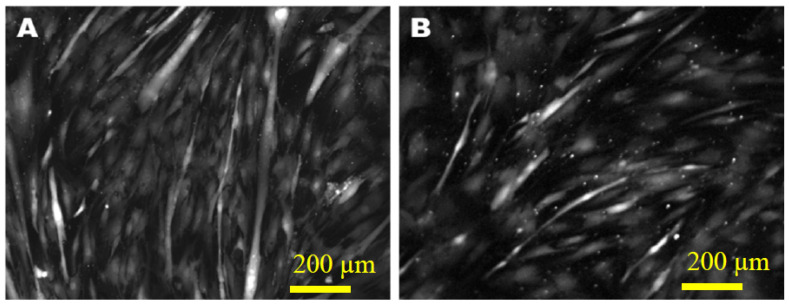
Myotubes were successfully isolated from patients at risk for MH. Myotubes with a negative CICR and no RYR1 or CACNA1S variant formed similar myotubes to CICR-positive with a known RYR1 mutation (**A**) Patient 1; (**B**) Patient 6).

**Figure 2 genes-14-02009-f002:**
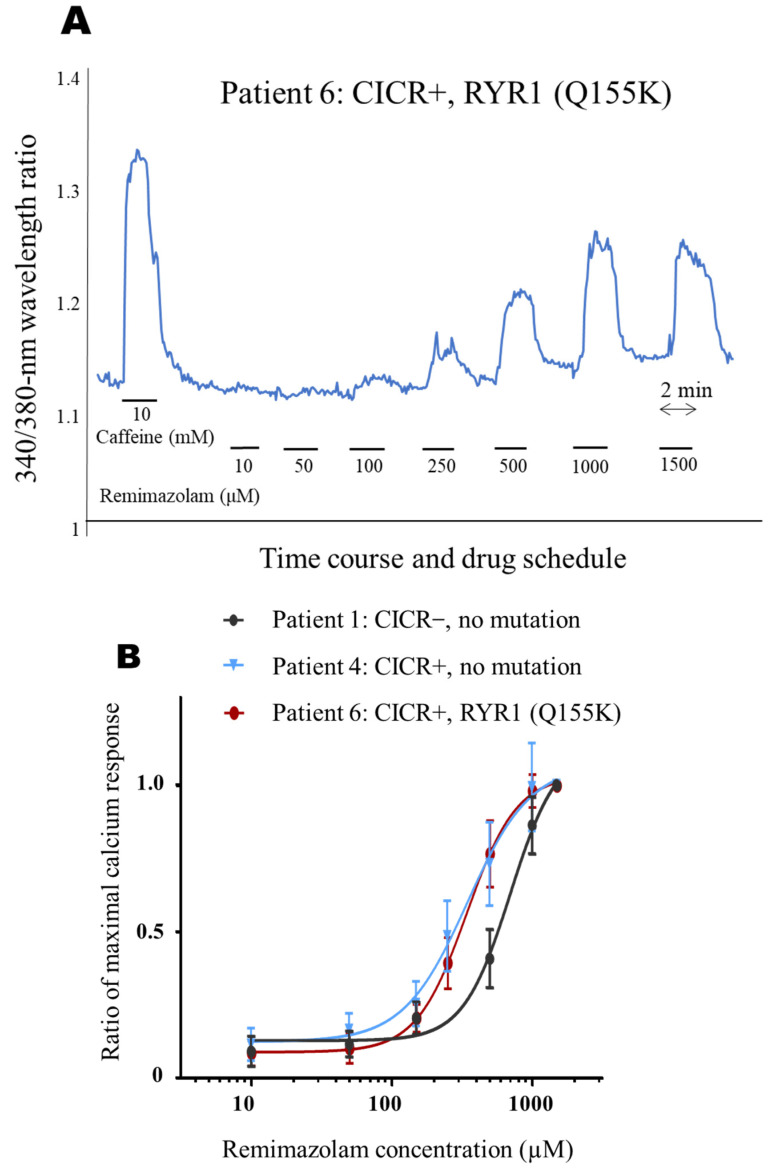
Dosing protocol for remimazolam. (**A**) Typical intracellular calcium concentration fluctuations during remimazolam dosing protocols (presenting data from Patient 6). The *X*-axis shows the time-course, and the *Y*-axis shows the 340/380-nm wavelength ratio. After confirming the response to caffeine, the cells were exposed to a stepwise increase in remimazolam concentration. (**B**–**K**) Concentration–response curves for remimazolam and intracellular calcium. (**B**) shows representative dose–response curves for three patient groups divided by CICR test results and the presence or absence of RYR1 variants. The myotube concentration–response curve for Patients 3–10 is more left-shifted than the myotube concentration–response curve for Patient 1 (Circle mark), indicating that lower concentrations of remimazolam increase intracellular calcium in CICR-positive myotubes.

**Table 1 genes-14-02009-t001:** Patient background.

Patient	Gender	Age	Reason for Taking CICR Test
1	F	24	relative has MH
2	F	80	suspicion of postoperative MH
3	M	62	relative has MH
4	M	23	develop MH
5	M	49	develop MH
6	F	73	relative has MH
7	M	38	relative has MH
8	F	14	relative has MH
9	M	25	develop MH
10	F	44	develop MH

The table shows the patients’ age, gender, and reason for CICR testing.

**Table 2 genes-14-02009-t002:** Genetic mutations and variants found in patients.

Gene	Amino Acid Change	Chromosome	Region	Type	Reference	Allele	Mutation Taster	CADD	REVEL	SIFT	PP2	VCEP Guidelines	EMHG Scoring Matrix
*RYR1*	Q155K	19	38934827	SNV	C	A	Deleterious	25.4	0.94	Deleterious (0)	Probably damaging (0.998)	Likely Pathogenic	VUS
	R163C	19	38934851	SNV	C	T	Deleterious	25.5	0.959	Deleterious (0)	Probably damaging (1.000)	Pathogenic	Pathogenic
	R2508H	19	38991539	SNV	G	A	Deleterious	25.4	0.898	Deleterious (0)	Probably damaging (0.987)	Likely Pathogenic	Pathogenic
	R4645Q	19	39062846	SNV	G	A	Deleterious	23.1	0.567	Tolerated (0.06)	Benign (0.007)	VUS	VUS
	W5020G	19	39078001	SNV	T	G	Deleterious	31	0.872	Deleterious (0)	Probably damaging (0.983)	VUS	VUS
*CACNA1S*	I639V	1	201044656	SNV	T	C	Benign	22.6	0.353	Tolerated (0.18)	Benign (0.022)	VUS	VUS

**Table 3 genes-14-02009-t003:** Results of CICR rate test and genetic testing of patients and EC_50_ of caffeine, 4CmC, and remimazolam.

Patient	CICR Test	Gene Mutation			EC_50_ (Mean ± SD)								
		*RYR1*	*CACNA1S*	*Stac3*	Caffeine (mM)	N	*p* Value	4-CmC (μM)	N	*p* Value	Remima (μM)	N	*p* Value
1	-	non	non	non	4.559 ± 1.017	25	-	279.1 ± 77.9	17	-	865.1 ± 317.8	9	-
2	-	non	I639V (Benign)	non	4.993 ± 1.317	7	0.8921	217.4 ± 80.6	14	0.0240	626.9 ± 117.4	6	0.0302
3	+	non	non	non	2.552 ± 1.083	11	<0.0001	130.7 ± 37.3	8	<0.0001	426.6 ± 40.3	5	<0.0001
4	+	non	non	non	3.244 ± 0.845	12	0.0005	163.7 ± 22.8	7	0.0001	405.2 ± 189.2	11	<0.0001
5	+	non	non	non	3.311 ± 0.746	17	0.0002	158.5 ± 39.5	7	<0.0001	208.0 ± 87.1	16	<0.0001
6	+	Q155K	non	non	2.753 ± 0.846	10	<0.0001	145.9 ± 59.7	9	<0.0001	396.6 ± 128.3	12	<0.0001
7	+	R4645Q	non	non	3.255 ± 0.775	9	0.0024	119.6 ± 57.6	11	<0.0001	206.0 ± 40.9	9	<0.0001
8	+	R2508H	non	non	2.782 ± 0.323	5	0.0029	107.2 ± 35.4	5	<0.0001	546.8 ± 71.5	4	0.0069
9	+	W5020G	non	non	1.680 ± 0.972	24	<0.0001	110.7 ± 42.1	30	<0.0001	339.8 ± 198.0	5	<0.0001
10	+	R163C	non	non	3.193 ± 0.681	26	<0.0001	142.5 ± 55.4	17	<0.0001	343.9 ± 119.0	14	<0.0001

The EC_50_ of each drug was compared between Patient 1 and the other patients. A *p*-value < 0.01 was considered significant. N: number of cells.

## Data Availability

The datasets used and/or analyzed during the current study are available from the corresponding author on reasonable request.

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
