# Peer review of "Effects of Remimazolam on Intracellular Calcium Dynamics in Myotubes Derived from Patients with Malignant Hyperthermia and Functional Analysis of Type 1 Ryanodine Receptor Gene Variants"

_genes, 2023, doi:10.3390/genes14112009_

Round 1

Reviewer 1 Report

Comments and Suggestions for Authors

Review

This is an interesting paper on the effects of Remimazolam on intracellular calcium concentration of myotubes derived from muscle biopsies of MH patients. The methods used are adequate and the results could have important human relevance but there are several issues which should be addressed.

Major points:

General comments: The Authors use the term EC50 incorrectly. The EC50 is the concentration of a drug that gives half-maximal response. Thus the sentence for example in row 19 is incorrect. It would sound like this: “The EC50 of caffeine, cresol, and remimazolam to induce intracellular calcium concentration change …”. There is a second generally used incorrect term in the same sentence: “intracellular calcium”. The Authors measured intracellular calcium concentration changes and not a Ca2+ in the intracellular solution. These should be corrected through the whole manuscript.

Introduction: The Authors talk about intravenous anesthetics in row 56-60 but forget to tell that the usual application of remimazolam is also intravenous. The Authors refer here two web pages (Ref. #10 and #11) but I did not find the cited information on these sites. Please give exact link to the desired information about safe intravenous anesthetics. The sentences in row 60-68 should be carefully rewritten. Reference #12 presents a case report of the successful usage of remimazolam. The Authors previous works showed that remimazolam did not act on wild type RyR1. Thus it should be strengthened here that the present work is concentrated on mutant calcium release channels.

Materials and Methods: My major criticism is connected to the cohort of patient. I do not believe that the Authors have only one absolute control (nor RyR1 neither CACNAS1 mutant) patient. All the results presented in the manuscript was compared to data obtained only one control patient. The Authors should be proved that the data they used can be accepted as control from a population (at least 3 patients). The muscle from which the biopsy was taken is missing. Reference #15 is not appropriate here because it also refers to a previous paper. Reference #16 is also not a good reference for CICR test since this is a review. The definition of MH predisposition should be clarified because CICR accelerated (MH) criteria was defined differently in reference #17. It should be clarified that the 3 drugs were tested on different fibers or on one fiber. In the latter case what was the order of the drugs and how much time was allowed to recover the fibers between drug applications? Do the N values in Table 3 present the number of fibers tested?

Results: The Authors show representative images from patient derived cultures in Figure 1. Did the Authors calculate anything (like differentiation index) from the cultures to prove the statement that they are similar? Unfortunately, the Authors forgot to place scale bars on the images. This deficiency must be filled. The age of cultures has also to be given in the legend.

The Authors present a representative fluctuations of intracellular calcium concentration during remimazolam dosing in Figure 2. It should be given in the legend from which patient’s data is shown. It would be helpful if the Authors show 3 representative curves from the 3 groups of patients they defined. The panel B-1 should be deleted since it is too crowded and panels B-2 – B-10 shows the individual pairs.

Discussion: It started with a sentence about Remifentanil. Is this equivalent to remimazolam? The sentence in row 279-280 should be deleted. The sentence in row 285-287 should be rewritten to underline that remimazolam has not effects on WILD TYPE RyR1. The Authors should give a clear hypothesis how the remimazolam can cause intracellular calcium concentration changes if RyR1 does not play role. The sentence in row 288-291 should be rewritten.

Patient #8 should be discussed because her results are almost identical to Patient #1.

Minor points:

Abstract: Please define EC50 and CACNA1S here.

Please move “agonist” after “(RyR1)” in row 15.

Materials and Methods: Please correct the text in row 83 and through the whole manuscript as follows: “Voltage-Gated Calcium Channel Subunit Alpha1”.

Please delete the first ”previously” in row 121.

Please delete “at 340/380-138 nm wavelength ratio” from 138-139.

Results: Please delete “respectively” from row 229.

Comments on the Quality of English Language

Some sentences should be rewritten. See my comments.

Author Response

Dear Reviewer

Thank you very much for peer review of our manuscript and providing important comments. And thank you for rating our study as interesting. We are thankful for the time and energy you expended. We carefully read the comments of you and made corrections based on the meaningful advice. We hope you will be satisfied with the revised manuscript and responses. 

Reviewer 2 Report

Comments and Suggestions for Authors

      The author investigated the intracellular calcium responsiveness of MH patient myotubes to caffeine, 4CmC, and remimazolam. Myotubes derived from patients with MH increased intracellular calcium levels at high concentrations of remimazolam, but clinically used concentrations of remimazolam did not increase intracellular calcium. The logic is clear. If the author can revise the following questions, it will be better.

Major concerns:

1.     For Result section, “3.6. Figures and Tables” should not be separated as a section and should be integrated into the other result sections;

2.     For figure legend or table notes, you use the full name of some terminology at the first time; in other cases, you can use abbreviations, such as “RYR1 and CACNA1S”

3.     For Figure2, it is better to use “A””B””C” rather than “B-1”B-2” et al.

4.     For Figure 2, please provide the statistical analysis, use “paired student-t test” at the same concentration.

Minor concerns:

1.     To ensure that the abstract is concise, it is better to cancel “One-way ANOVA with a Dunnett’s multiple comparisons test was used, and P<0.01 was considered significant” and “(P-values < 0.01)”.

2.     In line 46 of page2, please check the grammar “but avoiding exposure to predisposed patients is also”.

3.     Please provide the background of CLAS and RYR proteins in your introduction.

Comments on the Quality of English Language

Almost good, and just a minor errors in English grammars. 

Author Response

Dear Reviewer

Thank you very much for peer review of our manuscript and providing important comments. We are thankful for the time and energy you expended. We carefully read the comments of you and made corrections based on the meaningful advice. We hope you will be satisfied with the revised manuscript and responses.
